# Long-Term Effects of Polystyrene Nanoplastics in Human Intestinal Caco-2 Cells

**DOI:** 10.3390/biom11101442

**Published:** 2021-10-01

**Authors:** Josefa Domenech, Mariana de Britto, Antonia Velázquez, Susana Pastor, Alba Hernández, Ricard Marcos, Constanza Cortés

**Affiliations:** Grup de Mutagènesi, Departament de Genètica i de Microbiologia, Facultat de Biociències, Universitat Autònoma de Barcelona, Edifici Cn, Campus de Bellaterra, 08193 Cerdanyola del Vallès (Barcelona), Spain; josefa.domenech@uab.es (J.D.); mariana.wilson@hotmail.com (M.d.B.); antonia.velazquez@uab.es (A.V.); susana.pastor@uab.es (S.P.); alba.hernandez@uab.es (A.H.)

**Keywords:** nanoplastics, polystyrene nanoparticles, genotoxicity, intestinal barrier, Caco-2 cells

## Abstract

The increasing presence of micro- and nanoplastics (MNPLs) in the environment, and their consequent accumulation in trophic niches, could pose a potential health threat to humans, especially due to their chronic ingestion. In vitro studies using human cells are considered pertinent approaches to determine potential health risks to humans. Nevertheless, most of such studies have been conducted using short exposure times and high concentrations. Since human exposure to MNPLs is supposed to be chronic, there is a lack of information regarding the potential in vitro MNPLs effects under chronic exposure conditions. To this aim, we assessed the accumulation and potential outcomes of polystyrene nanoparticles (PSNPs), as a model of MNPLs, in undifferentiated Caco-2 cells (as models of cell target in ingestion exposures) under a relevant long-term exposure scenario, consisting of eight weeks of exposure to sub-toxic PSNPs concentrations. In such exposure conditions, culture-media was changed every 2–3 days to maintain constant exposure. The different analyzed endpoints were cytotoxicity, dysregulation of stress-related genes, genotoxicity, oxidative DNA damage, and intracellular ROS levels. These are endpoints that showed to be sensitive enough in different studies. The obtained results attest that PSNPs accumulate in the cells through time, inducing changes at the ultrastructural and molecular levels. Nevertheless, minor changes in the different evaluated genotoxicity-related biomarkers were observed. This would indicate that no DNA damage or oxidative stress is observed in the human intestinal Caco-2 cells after long-term exposure to PSNPs. This is the first study dealing with the long-term effects of PSNPs on human cultured cells.

## 1. Introduction

The exponential growth of plastic waste has been observed worldwide since the beginning of plastics’ large-scale production. Consequently, plastic waste has already become environmentally ubiquitous. Once in contact with diverse environmental factors, plastic items are increasingly fragmented and eventually generate micro- and nanoplastics (MNPLs), according to their reached sizes [1,2,3]. At the micro/nano range, the increase in the surface area-to-volume ratio of polymers may change the material’s reactivity and its potential toxicity. Although the MNPLs resulting from the degradation of plastic items (secondary MNPLs) represent a very important part of the environmental burden, there are MNPLs especially designed/produced at that size for different industrial purposes (primary MNPLs). Thus, the use of MNPLs beads in the production of cosmetics such as scrub and exfoliating products are continuously increasing, and finally, they end as plastic debris in the environment [4]. In addition, micro-/nanobeads of different plastics can also be useful for drug delivery [5].

Ingestion is considered one of the main routes for potential MNPLs human exposure, as it is the intake pathway for some of the more plausible sources of MNPLs such as contaminated food, liquids, and those initially entering through the respiratory system. In this regard, the experimental evidence of contamination of water and food sources with MNPLs is of particular concern for human health [6,7,8]. Although the hazard for human exposure to ingested MNPLs is potentially high, experimental data on the effects of this type of exposure is very limited. Apart from the observed effects of MNPLs ingestion in different species, mainly aquatic organisms, no direct evidence on humans exist, and only a few in vitro studies with human cell lines have been carried out to examine the cell internalization of MNPLs and the potentially harmful effects of MNPLs exposures [9,10,11]. It should be noted that the so-far published in vitro studies have used acute exposures and usually high concentrations of microplastic particles, as the exposure approach. This means that in vitro experimental data on the effects of chronic exposures are lacking. Consequently, there is an urgent need for new experimental data on the effects of nanoplastics exposure at lower—subtoxic—concentrations, and following long-term exposures lasting for weeks- to obtain more realistic estimates of the MNPLs-associated risk. Although the established in silico predictions state that chronic exposure to environmental concentrations of nanoplastics may cause genotoxicity, oxidative stress, and inflammation potentially leading to carcinogenic processes in a long-term human exposure scenario [12], experimental pieces of evidence in this regard are still lacking.

Therefore, the main objective of this study was to evaluate the effects of in vitro long-term exposures on human gastrointestinal cells. This type of cell system was selected assuming that ingestion is the main route of MNPLs intake in humans and, consequently, enterocytes became a relevant cell target, as they are the main components of the intestinal barrier. Our main focus was to observe the dynamics of polystyrene nanoplastics uptake over time, and to assess the potential cytotoxic and genotoxic effects that this exposure may induce. Consequently, we exposed Caco-2 cells, a broadly-used and well-established enterocytic cell line for toxicological studies, for eight consecutive weeks to polystyrene nanoparticles (PSNPs). To ensure a constant exposure condition, cell culture was replaced every 2–3 days with new media containing the desired concentration of PSNPs. These nanoplastics were chosen to extend our previous acute studies evaluating different biological endpoints [13,14]. Their internalization and accumulation were monitored throughout the study. Finally, different stress-related biomarkers were assessed at the end of the exposure period to evaluate the induction of potentially cytotoxic and genotoxic effects.

## 2. Materials and Methods

### 2.1. Cell Culture

Caco-2 human colon adenocarcinoma cells were maintained in Dulbecco’s modified Eagle’s High Glucose Medium (DMEM) without sodium pyruvate (Biowest, Nuaillé, France), supplemented with 10% fetal bovine serum, 1% non-essential amino-acids (Biowest, France), and 2.5 mg/mL Plasmocin (Invivo Gen, San Diego, CA, USA). Cells were kept in a humidified atmosphere of 5% CO_2_ at 37 °C and sub-cultured once a week into 25 cm^2^ dishes, according to the desired cell density. Cell growth was monitored daily and passaged at 80–90% confluence, to avoid differentiation in the cell monolayer. For the long-term experiments, the growth medium was changed every 2–3 days for a fresh medium with the treatment. The Caco-2 cell line was kindly provided by Dr. Isabella Angelis (Istituto Superiore di Sanità, Rome, Italy).

### 2.2. Nanoplastic Particles Characterization

Both the fluorescent (y-PSNPs) and non-fluorescent polystyrene nanoplastics (PSNPs) used in this study were commercially obtained (Spherotech, Inc., Chicago, IL, USA), having a nominal diameter of about 50 nm. To characterize these nanoplastics, nanoparticle dispersions were prepared at a concentration of 100 μg/mL in distilled water, and DMEM. To measure the average size of the nanoparticles, images were taken using a transmission electron microscopy (TEM) JEOL JEM-1400 instrument (Jeol LTD, Tokyo, Japan). The diameters of 100 randomly selected nanoparticles were measured with the Image J software (National Institutes of Health, Bethesda, MD, USA) and the mean size was calculated with GraphPad Prism 5 Software software (GraphPad Software, Inc., San Diego, CA, USA). Additionally, dynamic light scattering (DLS) and laser Doppler velocimetry (LDV) were used to measure the hydrodynamic size and the Z-potential of particles in water, and in DMEM at the same final concentration, in a Malvern Zetasizer Nano Zs zen3600 device (Malvern, UK).

### 2.3. Short-Term Exposure to Nanopolystyrene

The biological effects induced by PSNPs and y-PSNPs on Caco-2 cells were assessed after 24 h of exposure. To that purpose, 1.5 × 10^5^ cells were seeded in 12 well-plates and allowed to sit for 24 h. Thereafter, cells were exposed to the assayed concentrations of PSNPs or y-PSNPs for 24 h. Untreated cells were used as a negative control for all the experiments.

### 2.4. Nanopolystyrene’s Cytotoxicity Assessment

Acute potential PSNPs and y-PSNPs cytotoxic effects were evaluated to select suitable concentrations for the long-term exposure experiment. To this end, 2 × 10^5^ cells were seeded 24 h before the onset of the experiment, after which they were exposed to a wide range of different concentrations in triplicates: 0, 6.5, 13, 26, and 39 μg/cm^2^. After the exposure time, samples were washed twice with PBS 1x and trypsinized. The detached cells were diluted at 1:100 in Isoton and counted using a Beckman counter (Beckman Coulter, Brea, CA, USA). The average number of cells counted for each treatment group was compared to unexposed controls to calculate cell viability percentages.

### 2.5. Intracellular Nanopolystyrene Detection

TEM and confocal microscopy were used to determine PSNPs/y-PSNPs uptake by Caco-2 cells. For the TEM study, cells were exposed for 24 h to 0.26 and 6.5 μg/cm^2^ of PSNPs. After that, they were washed with PBS, trypsinized, pelleted, and fixed in 2.5% (v/v) glutaraldehyde (EM grade, Merck, Darmstadt, Germany) and 2% (*w/v*) paraformaldehyde (EMS, Hatfield, PA, USA) in 0.1 M cacodylate buffer (PB, Sigma-Aldrich, Steinheim, Germany) at pH of 7.4. Samples were then processed following conventional procedures, as previously described [15]. Images were taken using a Jeol 1400 TEM (Jeol LTD, Tokyo, Japan) equipped with a CCD GATAN ES1000W Erlangshen camera. Laser confocal microscopy was also used to assess the internalization of the y-PSNPs. To this aim, cells were seeded in Glass Bottom Microwell Dishes (MatTek, Ashland, MA, USA) and exposed to 0.26 and 6.5 μg/cm^2^ y-PSNPs concentrations for 24 h. After exposure, cells were stained for 15 min at room temperature. The cells’ nucleus was stained using Hoechst 33,342 (ThermoFisher Scientific, Waltham, MA, USA) and cells’ membranes were dyed using CellmaskTM Deep Red plasma (Life Technologies, Carlsbad, CA, USA) at 1:500 dilution in DMEM for both fluorophores. Cells were washed twice with DMEM after staining. Images of each sample were obtained using a Leica TCS SP5 confocal microscope and processed using Huygens essential 4.40p6 (Scientific Volume Imaging, Hilversum, The Netherlands). Unexposed cells were used as a negative control for both experimental approaches.

### 2.6. Long-Term Exposure to Nanopolystyrene

Caco-2 cells were exposed to four different non-cytotoxic concentrations of PSNPs and one concentration of y-PSNPs for 8 weeks. Nonspecific criteria exist to determine the length of the exposure. According to previous studies, exposures lasting for 4–10 weeks can be considered as chronic or long-term exposures [15,16] when in vitro treatments are used. The treatment concentrations were 0.0006, 0.26, 1.3 and 6.5 µg/cm^2^ for PSNPs exposures, and 0.26 µg/cm^2^ for y-PSNPs. The lowest treatment concentration would reflect an estimate of the human ingestion of 7 μg of plastic particles with the consumption of 225 g of mussels, according to the European Food Safety Authority [5], while the other concentrations were non-cytotoxic based on the cell viability assessment. Cells were routinely kept in 25 cm^2^ flasks throughout the exposure time-frame and amplified in 75 cm^2^ flasks for seeding the experiments conducted after 8 weeks of exposure. When using 75 cm^2^ flasks, cells were passaged at a density of 7 × 10^5^ cells per flask. When using 25 cm^2^ flasks, a seeding cell density of 2 × 10^5^ cells per replicate was used. Three replicates were maintained for each treatment concentration and for unexposed time-matched controls. It should be emphasized that exposures were done on undifferentiated Caco-2 cells. Although Caco-2 cells can be differentiated to enterocytes with functional polarity and tight-junction function, such a model of the intestinal barrier is not suitable for long-term exposures, such as those carried out in this study (eight weeks).

### 2.7. Nanopolystyrene Internalization

To study the dynamics of nanopolystyrene internalization throughout the long-term exposure, cells were exposed to 0.26 µg/cm^2^ of y-PSNPs for 8 weeks. Cell fluorescence in each replicate was measured with a cytometer (BD FACSCalibur, Becton Dickinson, Franklin Lakes, NJ, USA) after 24, 48, 72, and 96 h of exposure, and weekly thereafter. For cytometer measurements, 1 × 10^5^ cells of each replicate were diluted in 500 μL of PBS. Each sample was analyzed for the percentage of fluorescent cells present, as well as the relative fluorescence intensity of each sample. The experiment was carried out along with unexposed time-matched control cells.

### 2.8. Real-Time RT–PCR Gene Expression Analysis

The expression of the oxidative-damage (*HO1, SOD2, GSTP-1*) and general-stress (*HSP70*) related genes was analyzed by Real-Time RT–PCR after short and long-term exposure of Caco-2 cells to PSNPs. Additionally, *ACTB* was used as the housekeeping reference gene. Cells were exposed to PSNPs for 24 h (short term) or 8 weeks (long term). Both for short- and long-term exposed cells, RNA extraction was carried out using TRI Reagent^®^ (Invitrogen, Waltham, MA, USA), according to the product’s recommended protocol. Extracted RNA samples were then treated with RNase-free DNAse I (Turbo DNA-free kit; Invitrogen, USA) for 1 h and quantified using Nanodrop (Nanodrop Spectrophotometer ND-1000). Retrotranscription was carried out using 2000 ng of RNA per sample with the High-Capacity RNA-to-cDNA kit (Applied Biosystems, Bedford, MA, USA), and the amount of cDNA after retrotranscription was quantified in each sample, again using Nanodrop (Nanodrop Spectrophotometer ND-1000). Samples were then diluted in RNase-free water to achieve a final concentration of 10 ng/μL of cDNA. The real-time RT-PCR analysis was then conducted with the cDNA samples on a LightCycler-480 (Roche, Basel, Switzerland) to evaluate the expression levels of the targeted genes. Each 20 μL of reaction volume contained 5 μL of cDNA (50 ng of cDNA), 10 μL of 2× LightCycler-480 SYBR Green I Master (Roche, Switzerland), 3 μL of distilled water, and 1 μL of each primer (forward and reverse) at a final concentration of 10 μM. The primer sequences used are the following: *HO1* F: 5′-TCCGATGGGTCCTTACACTC-3′, R: 5′-AAGGAAGCCAGCCAAGAGA-3′; *GSTP1* F: 5’-CCAATACCATCCTGCGTCAC-3´, R: 5´-CAGCAAGTCCAGCAGGTTGT-3´; *HSP70* F: 5′-TGATCAACGACGGAGACAAG-3′, R: 5′-TCCTTCATCTTGGTCAGCAC-3′; *SOD2* F: 5´-GGCCTACGTGAACAACCTGA-3´, R: 5´-GAGCCTTGGACACCAACAGA-3´; *ACTB* F: 5´-GCATGGAGTCCTGTGGCATC-3´, R: 5´-CCACACGGAGTACTTGCGCT-3´. Three wells per replicate, concentration, and target gene were used. The LightCycler-480 parameters were as follows: pre-incubation at 95 °C for 5 min; 45 cycles of 95 °C for 10 s; 62 °C for 15 s; and 72 °C for 25 s. The data on the crossing points (Cp) for each sample was obtained using the LightCycler-480 software. Target gene values were normalized against the values for the housekeeping gene and analyzed statistically for significance.

### 2.9. Genotoxic and Oxidative DNA Damage Assessment in the Comet Assay

Genotoxic and oxidative DNA damage was evaluated in Caco-2 cells after 24 h and 8 weeks of exposure to different concentrations of PSNPs. Besides, negative and positive controls were set up. Positive controls were treated with 5 mM KBrO_3_, and 200 μM MMS for 30 min, as inducers of oxidative and genotoxic DNA damage, respectively. Exposed/control cells were centrifuged at 1000 rpm for 8 min and cell pellets were resuspended in PBS to achieve a dilution of 10^6^ cells per mL. Subsequently, each sample was mixed with previously heated agar, the mixture was dropped on pre-cooled GelBonds (GelBond^®^ film, GBF, Lonza, Bend, OR, USA) and they were left dry at 4 °C. All samples were dripped in two separate GBFs, one to assess oxidative DNA damage and the other for genotoxic damage. After drying, GBFs were submerged in lysis buffer (NaCl 2.5 M, EDTA 0.1 M, Tris 0.01 M, NaOH 0.2 M) and incubated overnight at 4 °C. The following day, GBFs were washed in enzyme buffer twice (HEPES 0.04 M, KCl 0.1 M, EDTA 0.0005 M, BSA 0.2 mg/mL) for 10 and 50 min. Samples were then incubated in enzyme buffer at 37 °C for 30 min, with the addition of formamidopyrimidine-DNA glycosylase (FPG) in the case of the GBFs used for oxidative damage analysis. Subsequently, GBFs were submerged in electrophoresis solution (NaOH 0.3M, EDTA 0.001 M) at 4 °C for 35 min and subjected to electrophoresis at 20 V and 300 mA for 20 min at 4 °C. Samples were then washed twice with PBS and once with water, and GBFs were fixed in pure ethanol for 1 h at room temperature. Ethanol was then removed and GBFs were air-dried. To dye samples, GBFs were submerged in SYBR Gold and left in agitation for 20 min. After that time, GBFs were rinsed with MilliQ water, mounted on slides, and visualized using an epifluorescence microscope (Olympus BX50F, Olympus Optical Co. Ltd., Tokyo, Japan). Comet counting and analysis were carried out using the Komet 5.5 software (Kinetic Imaging, Liverpool, UK). 100 nuclei per sample were counted. The software provided the percentages of DNA in comet tails for each of the counted nuclei. Oxidative DNA damage values were calculated by subtracting the percentages of total genotoxic damage per sample from the damage measured in samples treated with FPG.

### 2.10. Oxidative Stress Assessment with the DCFH-DA Method

Intracellular reactive oxygen species (ROS) production was evaluated after the exposure of Caco-2 cells to PSNPs for 24 h and 8 weeks. After the exposure time, cells were incubated with 20 µM dichloro-dihydro-fluorescein diacetate (DCFH-DA) in serum-free DMEM for 1 h at 37 °C. In both experimental approaches, positive control cells were treated with 100 mM H_2_O_2_ for 1 h before incubation with DCFH-DA. Cell fluorescence was then measured at 490/530 nm using the Victor 1420 Multilabel Counter fluorimeter (PerkinElmer, Waltham, MA, USA). For statistical analysis, the readings for each dose were averaged and normalized against the values for positive control samples.

### 2.11. Statistical Analysis

All experiments were carried out in triplicates and one-way ANOVA was carried out with the data from each of the experiments described above, to analyze their statistical significance, unless stated otherwise. To this end, GraphPad Prism 5 software (GraphPad Software, Inc., USA) was used. When convenient, Dunnett’s multiple comparison test was subsequently conducted. Statistical significance was set as * *p* ≤ 0.05, ** *p* ≤ 0.01, *** *p* ≤ 0.001.

## 3. Results

### 3.1. Nanoplastic Particles Characterization

The shape and size of PSNPs and y-PSNPs were assessed by TEM. As shown in Figure 1, both nanoparticles are round-shaped when diluted in distilled water or DMEM. Table 1 summarizes the results obtained for the nanoparticles’ characterization. TEM sizes were consistent with the ones indicated by the manufacturer, at around 50 nm diameter. However, the hydrodynamic radius, measured by DLS, showed larger particle sizes, especially for particles diluted in DMEM. The obtained polydispersity index (PdI) values indicate differences depending on the solvent used, showing a wider range of particle sizes when they are diluted in DMEM. The Z-potential values registered also showed differences in the aggregation state of particles depending on the solvent used. While dispersions in distilled water are stable, the ones in DMEM show a greater propensity to aggregation. This aggregation observed in DMEM, as confirmed by the PdI and Z-potential values, explain the variations in the DLS size between those PSNPs dispersed in water and in DMEM.

### 3.2. Short-Term PSNPs Cytotoxicity

Exposures lasting for 24 h were carried out at a concentration range of 0, 6.5, 13, 26, and 39 μg/cm^2^. Results indicate that the exposed cells displayed very low levels of cytotoxicity to PSNPs and y-PSNPs, as shown in Figure 2. Even at the highest 39 μg/cm^2^ concentration tested the cell viability remains very close to 100% after PSNPs and y-PSNPs exposures when compared to the untreated control. According to this, concentrations ranging from 0.006 to 6.5 μg/cm^2^ were selected for the assessment of PSNPs’ long-term effects. It should be remembered that we aimed to test “human realistic” exposure conditions, assuming exposures lasting for long-time to very low concentrations. Interestingly, the selected range includes a concentration resembling the potential exposure from food ingestion (0.0006 μg/cm^2^, equivalent to a potential exposure from a portion of mussels). The highest concentration used (6.5 μg/cm^2^) was the lowest tested to determine acute toxicity.

### 3.3. Short-Term PSNPs Internalization, Cellular Localization, and Subcellular Structural Effects

Confocal microscopy was used to analyze the ability of Caco-2 cells to internalize PS nanoplastic particles after short-term exposure. As seen in Figure 3, the particles were largely found inside the cells at all concentrations tested after 24 h of exposure, following an increased cellular accumulation trend at higher concentrations. Besides their location in the cytoplasm, y-PSNPs were also present in the cell nuclei at all concentrations. The uptake of PSNPs was also assessed by TEM. As shown in Figure 4, PSNPs-exposed Caco-2 cells exhibited a considerable number of intracellular particles. The untreated control group displayed normal cell morphology, with a well-organized nucleus and nucleolus, mitochondria with regular cristae, and normal cellular membranes. On the other hand, PSNPs-exposed samples showed structural differences at every concentration tested, which formed dark electron-dense structures in the perinuclear region (indicated with yellow arrows in Figure 4). We also observed, as pointed out with green arrows in Figure 4, an increased accumulation of electron-dense vacuoles and lysosomes at the 6.5 μg/cm^2^ concentration. The images also showed mitochondrial cristae swelling for the highest concentration (indicated with orange arrows in Figure 4). Taken together, these results evidence that PSNPs uptake causes subcellular responses that increase in a dose-dependent manner. Given these results, we chose to continue the long-term experiments with PSNPs using 6.5, 0.26, 1.3, and 0.0006 μg/cm^2^.

### 3.4. Long-Term y-PSNPs Internalization

We assessed the internalization of y-PSNPs by flow cytometry at the concentration of 0.26 μg/cm^2^ during the experiment lasting for eight weeks. As shown in Figure 5A, approximately 20% of the exposed cells had detectable internalization levels of y-PSNPs. This proportion remained stable throughout the following weekly experimental measurements. On the other hand, the accumulation of y-PSNPs in cells that presented fluorescence stabilized after two weeks of treatment, with fluorescent cells displaying a relative fluorescence intensity three times higher than the untreated control, as shown in Figure 5B.

### 3.5. Gene Expression Analysis

We analyzed the gene-expression pattern of different oxidative and general stress-related genes after 24 h and 8 weeks of PSNPs exposure to determine whether PSNPs can induce stress-related responses in Caco-2 cells. The selected target genes were *HO1*, *SOD2*, *GSTP1,* and *HSP70*. On the one hand, as shown in Figure 6, we could not find any significant change in the expression of the analyzed genes after 24 h of exposure to PSNPs despite a slightly increasing tendency as the concentration of PSNPs increased. Nevertheless, the effects gained statistical significance after the long-term exposure of the cells to PSNPs. Thus, transcriptional expression levels of *HO1* and *SOD2* showed a significant increase, when compared to untreated samples at practically all the tested concentrations. *GSTP1* and *HSP70*, on the other hand, did not show any relevant variations when compared to the untreated control. Taking together, these results indicate that PSNPs significantly alter the oxidative stress-related genes’ expression under long-term exposure regimes.

### 3.6. Genotoxic and Oxidative DNA Damage

The potential genotoxic effect of short- and long-term exposures to PSNPs was examined using the comet assay to detect single and double-strand breaks, as well as oxidative DNA damage. The comet assay revealed low levels of genotoxic and oxidative DNA damage, both for cells exposed to PSNPs for 24 h and 8 weeks. Only the 0.26 μg/cm^2^-treated sample after 8 weeks of exposure showed a significant increase in the genotoxic damage observed when compared to the untreated cells (Figure 7A). The slight variations in the genotoxic damage observed after the short-term exposure were not significantly different from those seen in the control group. As for the oxidative DNA damage (Figure 7B), Caco-2 cells exposed for 24 h to the highest PSNPs concentration presented an increased level of damage when compared to that found in negative control samples. However, these variations did not attain statistical significance. Summarizing, these results show that cells exposed to PSNPs for both 24 h and 8 weeks do not increase their levels of genotoxic and oxidative DNA damage.

### 3.7. Intracellular ROS Production

We assessed the intracellular levels of ROS production with the DCFH-DA detection assay. The results show that PSNPs did not induce statistically significant differences in ROS levels when compared to untreated controls, neither after 24 h or 8 weeks of exposure (Figure 8). Conversely, relevant oxidative damage can be detected in this cell line, as seen by the increase in fluorescence in the positive control cells treated with H_2_O_2_. Thus, these results suggest that PSNPs exposure did not cause an increase of oxidative stress in the PSNPs-exposed samples.

## 4. Discussion

Given the increasing presence of plastic waste’s derivatives such as MNPLs in the natural environment, substantial efforts are required to evaluate their potential hazard and associated risks for humans. Unfortunately, the number of biomonitoring studies or studies involving in vivo models is scarce, and most literature focuses on in vitro cell-line experimental models to assess the potential risk of nanoplastics [17]. In such toxicological studies, PSNPs are the most widely used MNPLs as PS is a highly abundant plastic polymer in the environment, and representative PSNPs are commercially available. However, the reports on the effects of this nanomaterial are conflicting. On the one hand, numerous reports claim that no harmful effects are observed in undifferentiated and differentiated intestinal cells [8,13,18,19,20]. On the other hand, toxic effects have been reported in blood, brain, epithelial, and placental human cells [19,20,21]. These differences might be due to the inherent characteristics of the cell model used. For instance, our group used different white blood cell types to assess the effects of ex vivo exposures on several toxicity biomarkers, and the results show that, albeit all the cell lineages presented toxic effects, there were substantial differences among each target cell type [14]. Nonetheless, all these reports evaluate the effects of nanoplastic on a short time window, while adverse effects on human health could also rise due to the accumulation of nanoplastics following continuous exposure.

One of the primary exposure routes of MNPLs is ingestion. Accordingly, we selected Caco-2 cells, a well-known and established intestinal model for in vitro nanotoxicology studies [22,23]. Our results show that the viability of Caco-2 cells acutely exposed to various concentrations of PSNPs and y-PSNPs remained stable. This suggests that this cell line may be particularly resistant to the PS nanoparticles’ cytotoxic effects, as decreased viability after PSNPs exposure has been observed in other human cell lines [24,25]. Furthermore, previous studies have also observed no significant cytotoxic effects in Caco-2 cells exposed to PSNPs, either in their differentiated or undifferentiated state [13,26]. Thus, it is possible that due to its function as a primary barrier in the human body, intestinal cells show higher resilience to PSNPs’ potential cytotoxic effects. To determine if the observed resistance of Caco-2 cells is due to their particular characteristics, or because they are members of the intestinal barrier, other intestinal cells, such as i.e., HT-29 should be used following the same procedures used with the Caco-2 cells.

Despite the non-toxic effects of PS nanoparticles, they internalize efficiently in Caco-2 cells. The internalization of y-PSNPs after 24 h was widespread, with fluorescent particles identified inside the exposed cells and their nuclei even for the lowest exposure concentration. The uptake of PSNPs by Caco-2 cells was deeply evaluated in a previous study using flow-cytometry and TEM, in addition to confocal technology [13]. Thus, taking all the data together, we can confirm that y-PSNPs are easily accumulation in exposed cells, and it is done in a concentration-dependent manner. These findings suggest that PS nanoplastics can enter exposed cells and reach the nucleus, potentially inflicting structural or genotoxic damage on exposed cells. In fact, at higher concentrations of PSNPs exposure, some ultrastructural alterations in mitochondria were evident, suggesting that PSNPs exposure could cause organelles’ dysfunction. These observations are in line with previous studies, which have recorded the internalization of nanoplastic particles and subsequent accumulation in lysosomes [24,27]. One such study found that PS nanoplastic internalization increases linearly over time, with nanoplastic particles irreversibly stored in lysosomes once inside exposed cells [27]. Additionally, the other study found that amine-modified PSNPs caused alterations to cells’ lysosomes, ultimately leading to an increased generation of ROS, mitochondrial dysfunction, and subsequent activation of the apoptosis pathway [24]. However, our interest focused on the accumulation dynamics of PSNPs during long-term exposure, which would mimic the oral intake by ingestion. Fluorescence measurements throughout eight weeks of exposure to y-PSNPs revealed that, at the lowest concentration where we could detect a fluorescent signal (0.26 µg/cm^2^), 20% of the exposed cells had internalized the y-PSNPs after 48 h, and this proportion was maintained for the rest of the exposure time. These results are in line with those of previous studies which have recorded in vitro internalization of nanoplastics by human cells [21,25,28,29,30,31]. In particular, the internalization of PSNPs of two different sizes by human gastric adenocarcinoma cells was studied [25]. Their results showed that both PSNPs were readily internalized by exposed cells, reaching saturation after 1 h of treatment. Furthermore, they found evidence that internalization occurred thanks to an energy-dependent process rather than diffusion through cell membranes and deduced that a release process could be activated upon reaching internalization saturation. While the y-PSNPs of the 0.26 µg/cm^2^ concentration used in our study were not internalized by 100% of the exposed cells, a similar release process could be responsible for maintaining levels of internalization relatively stable throughout the eight-week exposure. Additionally, the accumulation of y-PSNPs continued to increase until stabilization at two weeks of exposure. Previously published data analyzing PSNPs accumulation dynamics have shown that amine-modified PSNPs accumulation was observed in the lysosomes of exposed human astrocytoma cells, leading to lysosomal swelling and, ultimately, apoptosis [24]. It must be noted that this accumulation dynamic was evaluated only during 24 h. While no such effects were evaluated in the long-term exposure conducted in this study, the mechanisms of accumulation of PSNPs in Caco-2 cells’ lysosomes cannot be discarded.

One of the main risks of chronic human exposure to non-cytotoxic concentrations of environmental MNPLs is the potential induction of effects associated with cell transformation and the initiation of the carcinogenic process. In this regard, many early hallmarks of carcinogenesis have been described, such as a higher incidence of DNA damage and an increase in oxidative stress [32]. In the current study, none of these significant transforming effects associated with the long-term exposure of human Caco-2 cells PSNPs was observed.

Several stress-related genes have been associated with transformation. The *HO1* gene codes for heme-oxygenase enzyme 1, which mediates the first step of heme metabolism. This enzyme has cytoprotective and anti-inflammatory properties, which may respond to a variety of stimuli, including hypoxia and oxidative stress [33]. The *SOD2* gene, on the other hand, encodes superoxide dismutase 2, a mitochondrial enzyme that removes superoxide originated from oxidative phosphorylation, protecting the cells from reactive oxygen species. Thus, *SOD2* plays a role in the protection against oxidative stress, and its dysfunction has been associated with several diseases involving mitochondrial dysfunction [34]. *GSTP-1* belongs to a gene family encoding glutathione S-transferases, involved in different cell detoxification pathways by catalyzing the conjugation of hydrophobic and electrophilic compounds with reduced glutathione. These genes are upregulated in response to oxidative stress and are overexpressed in certain tumors [35]. Finally, the *HSP70* gene codes for heat shock proteins, which offer protection against heat or chemical stress, by assisting in the refolding of denatured peptides, avoiding proteolytic degradation [36]. Gene expression for *HO1, SOD2, GSTP1,* and *HSP70* showed no significant changes after short-term exposure to PSNPs. As these genes have protective functions against oxidative and chemical-induced stress, their expression is expected to increase in samples exposed to hazardous agents, and overexpression of the *GSTP1*, *HO1,* and *HSP70* genes have been associated with the increased survival of transformed cells [33,35,37]. In contrast, the current long-term study found significant changes in the expression levels of *HO1* and *SOD2* genes, suggesting that chronic exposure to non-cytotoxic doses of PSNPs increases the stress-related responses of the exposed cells, and therefore it could induce stress-related carcinogenic effects at the studied endpoint.

Another important toxicological endpoint assessed was genotoxicity, which is regularly used as a surrogate biomarker for genetic-associated pathologies such as cancer [38]. Furthermore, the assessment of the genotoxic potential is required for all new chemical substances given the impact on public health that these compounds could have. Therefore, the genotoxicity data reported in this work provide relevant information to the hazard assessment of MNPL exposure. Our study did not find any relevant changes in genotoxic and oxidative DNA damage in cells exposed to PSNPs for 24 h or 8 weeks. While some previous studies have found higher levels of DNA damage in samples treated with PSNPs, others have recorded no genotoxic or oxidative DNA damage associated with PSNPs exposure. On the one hand, one study observed DNA damage in half of the lymphocytes treated with PSNPs after acute exposure, while another one shows that PSNPs’ genotoxic damage depends on the white cell lineage analyzed [14,39]. On the other hand, another study reported differences in the genotoxic effects observed in human lung cancer and macrophage cells after acute exposure to differently functionalized PSNPs particles [40]. Interestingly, these authors found no significant genotoxic effects of exposure to pristine, non-functionalized PSNPs. These differences highlight the importance of particle structure in the observed effects, both at acute and long-term endpoints.

As for the levels of ROS present in exposed cells, many previous studies have documented increases in oxidative stress due to nanoplastics exposure [21,24,28,40]. However, we did not observe such an effect in our study. Instead, exposed samples showed similar levels of ROS to unexposed controls both after the short and long-term exposure, indicating that exposure to pristine PSNPs did not cause an increase in the oxidative stress levels of exposed cells.

Overall, the only indication of significant stress-related pathways observed was the overexpression of *HO1* and *SOD2* genes in Caco-2 cells exposed to PSNPs for eight weeks. However, we did not found other evidence of ROS production, DNA damage, or oxidative damage throughout all measured endpoints, suggesting that long-term exposure to pristine PSNPs alone does not induce any of these effects in Caco-2 cells. However, environmental weathering of nanoplastic particles may cause changes to their properties that could alter their toxicity towards exposed organisms, such as size and surface charge [1,31]. Additionally, the structural change could alter their adsorption of environmental contaminants, which is already well documented and may induce indirect harmful effects on exposed organisms, especially if accumulated and amplified throughout the food web [12,41,42]. Thus, it is important to obtain experimental data that better reflects the realistic conditions of exposure to nanoplastics, as toxicities observed in controlled laboratory experiments, such as those focusing on acute exposures, may vastly differ from the real hazard posed by the chronic exposure to weathered and altered nanoplastics. Our study, which proposes a long-term exposure at lower concentrations, aims to better reflect environmental conditions, and thus give information that is more alike to the real-case scenario.

## Figures and Tables

**Figure 1 biomolecules-11-01442-f001:**
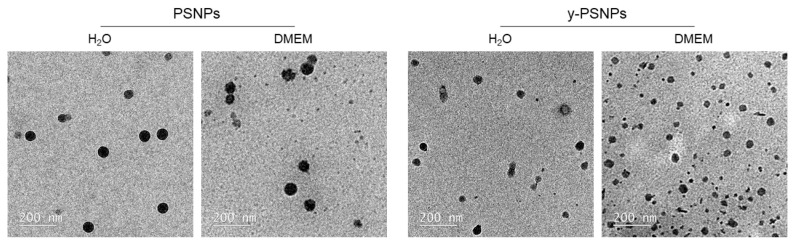
Representative TEM images of PS nanoparticles (PSNPs and y-PSNPs). Samples were prepared using 26 μg/cm^2^ dilutions, in distilled water and DMEM, of each nanomaterial.

**Figure 2 biomolecules-11-01442-f002:**
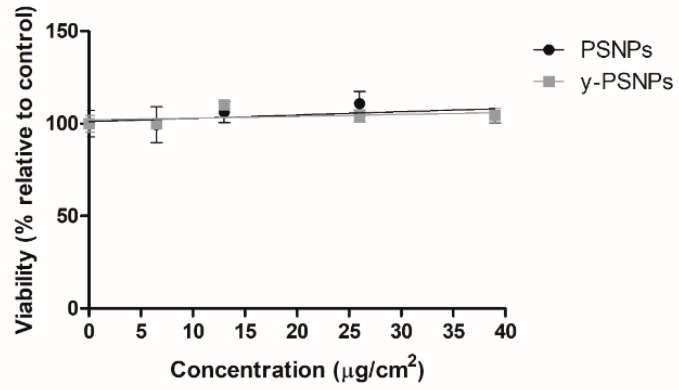
Relative survival of Caco-2 cells after 24 h of exposure to PSNPs and y-PSNPs at concentrations ranging from 0 to 39 μg/cm^2^. Data are presented as the percentage of counted cells relative to the untreated control ± SEM.

**Figure 3 biomolecules-11-01442-f003:**
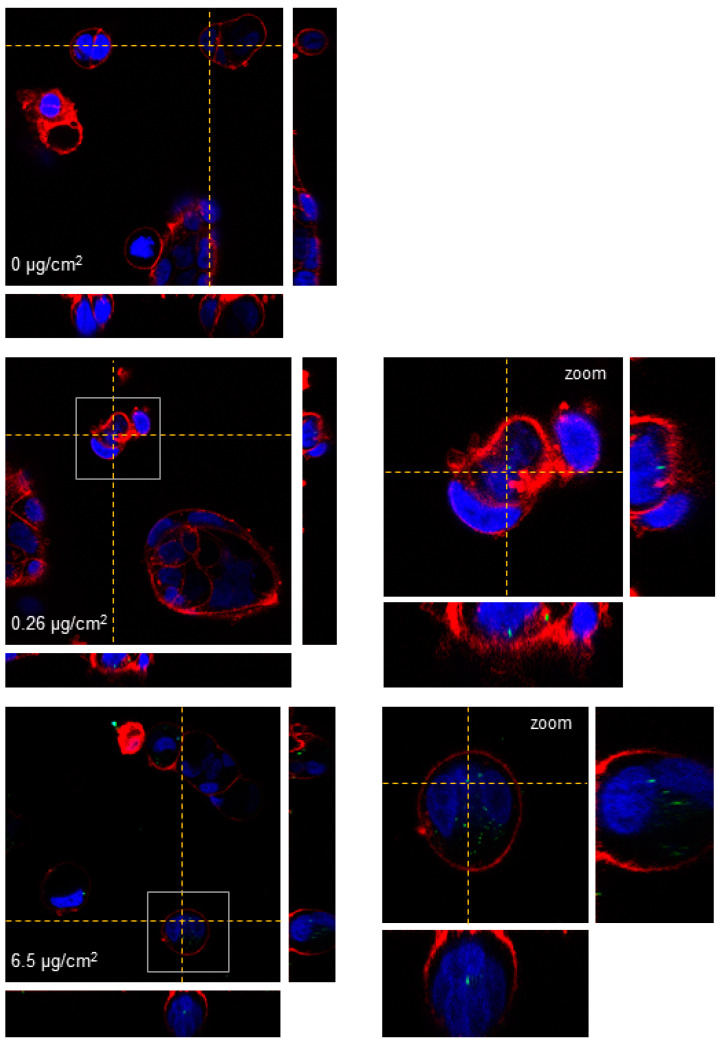
Three-dimensional confocal microscopy images of undifferentiated Caco-2 cells after a 24 h exposure to y-PSNPs. Nuclei are stained in blue, cell membranes in red, and nanoparticles are depicted in green. Dotted lines point out the plane from where the orthogonal views are projected. Images on the left correspond to cells exposed to the different concentrations of y-PSNPs indicated while images on the right correspond to the zoomed area highlighted by a grey square.

**Figure 4 biomolecules-11-01442-f004:**
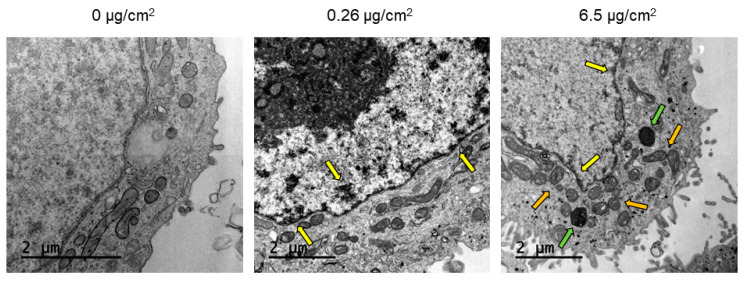
TEM images of Caco-2 cells after 24 h of exposure to increasing concentrations of PSNPs. Dark and electron-dense formations observed in the perinuclear region are pointed out with yellow arrows, while PSNPs accumulations in vacuoles and lysosomes are indicated with green arrows. Orange arrows point out the induction of mitochondrial cristae swelling.

**Figure 5 biomolecules-11-01442-f005:**
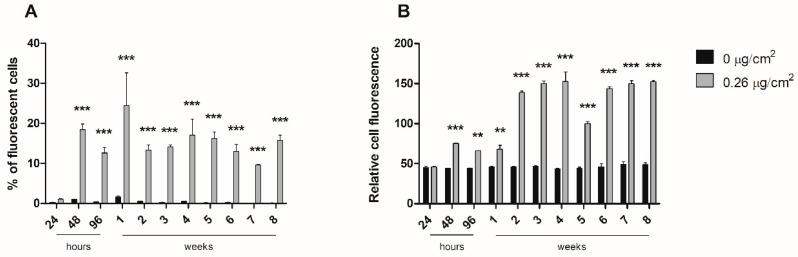
y-PSNPs internalization and accumulation in Caco-2 cells throughout eight weeks of exposure to 0.26 μg/cm^2^. Percentage of fluorescent Caco-2 cells (**A**), indicating y-PSNPs internalization, and relative fluorescence intensity of cells, compared to untreated cells (**B**), indicating y-PSNPs accumulation, at 24, 48, and 96 h and weekly are shown. Values are plotted as means for all three replicates and SEM of each treatment group. Statistical significance was determined by two-way ANOVA. ** *p* ≤ 0.01, *** *p* ≤ 0.001.

**Figure 6 biomolecules-11-01442-f006:**
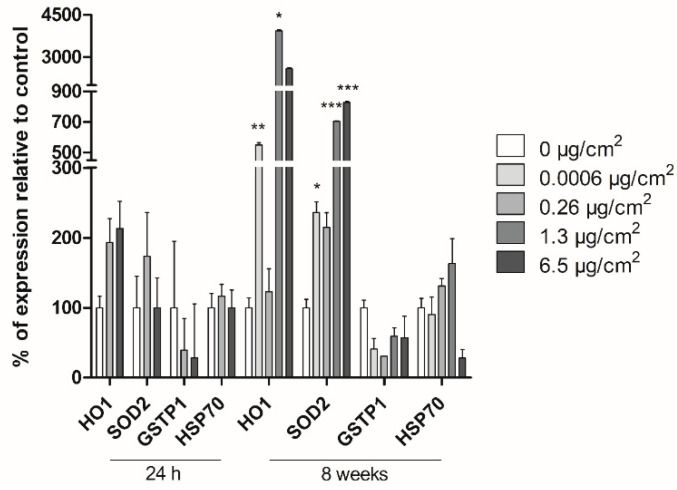
Study of Caco-2 cells’ response after 24 h and 8 weeks of PSNPs exposure using Real-Time RT-PCR. The percentage of expression for each gene is shown compared to untreated controls, per dose and gene. Data are presented as mean ± SEM for each treatment dose and analyzed by the student’s *t*-test. * *p* ≤ 0.05, ** *p* ≤ 0.01, *** *p* ≤ 0.001.

**Figure 7 biomolecules-11-01442-f007:**
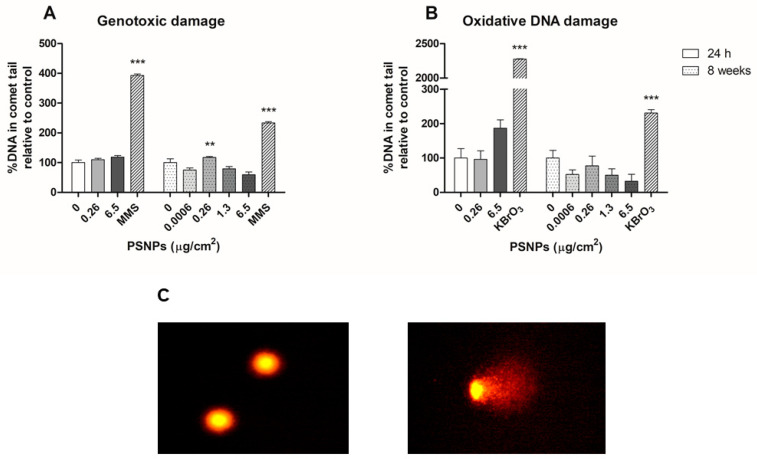
Genotoxic (**A**) and oxidative (**B**) DNA damage in Caco-2 cells after 24 h and 8 weeks of PSNPs exposure, as evidenced by comet assay. Data represent the percentage of DNA in comet tails, relative to that of unexposed controls as mean ± SEM. Statistical significance was determined by one-way ANOVA with Dunnett’s multiple comparison post-test. Comet figures (**C**) of undamaged cells (left) and damaged cell (right). ** *p* ≤ 0.01, *** *p* ≤ 0.001.

**Figure 8 biomolecules-11-01442-f008:**
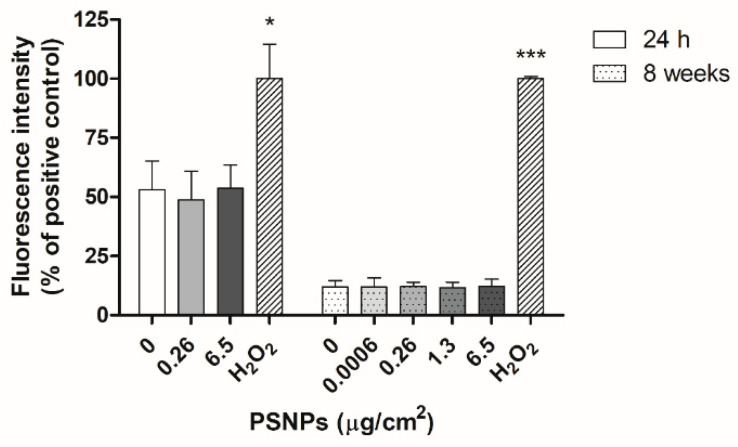
Presence of intracellular ROS levels as detected by DCFH-DA fluorescence assay in cells exposed to PSNPs for 24 h and 8 weeks, untreated controls, and positive controls treated with H_2_O_2_. The percentage of fluorescence intensity relative to the positive control is shown. Data are represented as mean ± SEM for each exposure concentration and analyzed by one-way ANOVA with Dunnett’s multiple comparison post-test. * *p* ≤ 0.05, *** *p* ≤ 0.001.

**Table 1 biomolecules-11-01442-t001:** PS nanoparticles parameters characterized by TEM and Zetasizer Nano ZS.

	PSNPs		y-PSNPs	
Dispersant	H_2_O	DMEM	H_2_O	DMEM
Size (nm) (TEM)	52.99 ± 14.68	48.59 ± 16.38	44.19 ± 28.54	55.21 ± 12.76
Size (nm) (DLS)	86.33 ± 10.20	158.28 ± 10.85	112.87 ± 3.11	377.52 ± 43.05
PdI (DLS)	0.10 ± 0.09	0.44 ± 0.09	0.35 ± 0.02	0.60 ± 0.06
Z-potential (mV) (LDV)	−36.00 ± 7.88	−9.31 ± 0.67	−45.97 ± 3.84	−9.80 ± 0.33

## Data Availability

Not applicable.

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
