# Peer review of "Long-Term Effects of Polystyrene Nanoplastics in Human Intestinal Caco-2 Cells"

_biomolecules, 2021, doi:10.3390/biom11101442_

Round 1

Reviewer 1 Report

The research proposed by Domenech and Co-authors deals about the long-term effects of nanoplastics in a cell line used as a model of human enterocytes. This type of research, aimed to establish negative effects of human exposure to environmental nanoplastics, is very actual and interesting. Authors describe their work in a fine and appealing way; however, I have to highlight some weaknesses in their approach which make results and/or their presentation not completely convincing.

Major points

Cytotoxicity assessment has been done only by count detached cells; maybe some other technique could be give more accurate results. Nanoparticles concentration over 39 microg/cm2 could also be tested. No information has been given about cytotoxicity after prolonged treatments. Intracellular nanoparticles detection showed that the most of the fluorescence began to be detected after 48 h of treatments, Thus, cytotoxicity and analysis of structural modification should be also evaluated later than 24 h.

To my experience to evaluate gene expression, it is essential also to monitor protein levels. Authors only reported mRNA analysis and they didn’t show protein detection at any time. Thus, data on changes in expression level of stress genes are weak.

Minor points

Representative images of comet assay should be added.

Reviewer 2 Report

Dear authors,
there are several inconsistencies in the work.
it is necessary to argue what caused the considerable difference in size in the DMEM medium, which you found from the DLS analysis.
In figure 1 the TEM images do not seem to match the information reported in table 1.
Figure 4 needs to be improved as the information reported about the mitochondria cannot be detected from the images shown; furthermore, the text refers to the internalization of nanoparticles at all concentrations so it would be advisable to insert images relating to all concentrations.
fluorescence images do not show the same internalization rate as reported in electron microscopy images.
The work therefore requires a more in-depth analysis at the uptake level.

Reviewer 3 Report

Review Report

For Article Long-term effects of polystyrene nanoplastics in human intestinal Caco-2 cells

In connection with the ever-increasing production and consumption of plastic in the modern world, it is relevant to study the effect of micro- and nanoplastics on human health. Most of the publications in this area are devoted to the study of the toxicity of plastics for animals, only a few in vitro studies with human cell lines fewer works on human tissues, but among them there are the lack of works on long-term exposure. This work is first study dealing with the long-term effects of polysterene nanoparticles (PSNPs) on human cultured intestinal Caco-2 cells for duration longer 1 month. Seven methods were used to assess the toxicity of particles. The obtained results attest that PSNPs accumulate in the cells through time, inducing changes at the ultrastructural and molecular levels and increased cell and oxidative gene expression of intestinal cells. Nevertheless, no DNA damage or oxidative stress is observed. The conclusions are significant. The hypothesis of the difference between long-term exposure of low concentration of PSNPs and short-term exposure is confirmed. The results provide an advance in current knowledge about accessing risk of micro- and nanoplastics to humans.

The introduction provides sufficient background and includes relevant references. The article is written in an appropriate way. The study is correctly designed and technically sounds. Main way of intake of these particles is gastrointestinal, that why the culture of intestinal Caco-2 cells was selected for the study. Choice of PSNPs is due a highly content polysterene in the environment, and the commercial availability of representative PSNPs. Variety of methods for assessing the toxicity of particles is present in the study. The methods, tools, software, and reagents are described with sufficient details to allow another researcher to reproduce the results.  There are sufficient repetitions and presence of controls.  The data and analyses are presented appropriately. Table, figures and figure captions are meaningful and understandable. The results are interpreted appropriately. The data are robust enough to draw the conclusions. The conclusions are justified and supported by the results conclusions are supported by the results. There are no inappropriate self-citations

The conclusions are interesting for the readership of the Journal because PSNPs are potential bioactive substances, whose action is of interest to molecular medicine (toxicology). The paper will also attract a wide readership in connection with the popularization of the topic of plastic pollution.

The investigated problem is not studied appropriately yet and requires further close study, however, this work will contribute to understanding and further advance in the field of studying the safety of nanoplastics. The undoubted advantage of this study is long-term observation of PSNPs action which revealed the features of chronic exposure. The study poses future challenges related to the observation of other tissues and using a different type of plastic particles (as close as possible to real conditions).

Comments

I would also recommend mentioning the following review to the introduction (Heddagaard and Moller, 2020).

Duration of the experiment does not correspondence fully to the definition of long-term (chronic) exposure). As usual it takes several months.

The choice of exposure duration 8 weeks is not explained.

Experimental concentration choice is not clear from text:

  1. 254-258 “Even at the highest 39 μg/cm2 concentration tested the cell viability remains very close to 100% after PSNPs and y-PSNPs exposures when compared to the untreated control. According to this, concentrations ranging from 0.006 to 6.5 μg/cm2 were selected for the assessment of PSNPs' long-term effects”.

It is not clearly understood wherefore differentiated to enterocytes Caco-2 cells are not suitable for long-term exposures:

  1. 150-152 “Although Caco-2 cells can be differentiated to enterocytes with functional polarity and tight-junction function, such a model of the intestinal barrier is not suitable for long-term exposures, such as those carried out in this study (eight weeks).”

It is useful to note that PSNP are interesting not only from negative point of view of contamination, but for drug delivery.

It is useful to add in discussion that used one cell line culture model is not fully adequate object for modeling intestinal consumption because there is triple culture model of the intestinal mucosa (Schimpel et al., 2014).

There are minor inaccuracies:

  1. 22 instead of “culture-media was removed every 2-3 days” it would be more appropriate “culture-media was changed every 2-3 days”
  2. 89 “every 2/3 days” need to be changed to “every 2-3 days”
  3. 99-101 “Additionally, dynamic light scattering (DLS) and laser Doppler velocimetry (LDV) were used to measure the Z-potential and the hydrodynamic size of particles in water” need to change the order of words in a sentence: “Additionally, dynamic light scattering (DLS) and laser Doppler velocimetry (LDV) were used to measure the hydrodynamic size and Z-potential the of particles in water”
  4. 115 “diluted 1:100” need to add a preposition “diluted at 1:100”

Overall Recommendation: Accept after Minor Revisions

Heddagaard, F. E. and P. Moller 2020. Hazard assessment of small-size plastic particles: is the conceptual framework of particle toxicology useful? Food and Chemical Toxicology 136: 14. doi: 10.1016/j.fct.2019.111106

Schimpel, C., Teubl, B., Absenger, M., Meindl, C., Fröhlich, E., Leitinger, G. and Z.E. 2014. Development of an Advanced Intestinal in Vitro Triple Culture Permeability Model To Study Transport of Nanoparticles Mol. Pharmaceutics 11: 3

Round 2

Reviewer 1 Report

I approve the manuscript in the present form.

Reviewer 2 Report

Dear authors
the answers sent are not exhaustive, the images, although representative, must be able to make the data understood. Furthermore, I continue to find a considerable difference in information between the different experiments.